# Inertial Sensor-Based Quantification of Movement Symmetry in Trotting Warmblood Show-Jumping Horses after “Limb-by-Limb” Re-Shoeing of Forelimbs with Rolled Rocker Shoes

**DOI:** 10.3390/s24154848

**Published:** 2024-07-25

**Authors:** Craig Bark, Patrick Reilly, Renate Weller, Thilo Pfau

**Affiliations:** 1The Royal Veterinary College, University of London, London NW1 0TU, UK; 2Department of Clinical Studies, New Bolton Center, University of Pennsylvania, Philadelphia, PA 19348, USA; 3Faculty of Veterinary Medicine, University of Calgary, Calgary, AB T2N 1N4, Canada; 4Faculty of Kinesiology, University of Calgary, Calgary, AB T2N 1N4, Canada

**Keywords:** horse, shoeing, inertial sensor, movement symmetry, push-off, weight bearing, straight line, lunge exercise

## Abstract

Hoof care providers are pivotal for implementing biomechanical optimizations of the musculoskeletal system in the horse. Regular visits allow for the collection of longitudinal, quantitative information (“normal ranges”). Changes in movement symmetry, e.g., after shoeing, are indicative of alterations in weight-bearing and push-off force production. Ten Warmblood show jumping horses (7–13 years; 7 geldings, 3 mares) underwent forelimb re-shoeing with rolled rocker shoes, one limb at a time (“limb-by-limb”). Movement symmetry was measured with inertial sensors attached to the head, withers, and pelvis during straight-line trot and lunging. Normalized differences pre/post re-shoeing were compared to published test–retest repeatability values. Mixed-model analysis with random factors horse and limb within horse and fixed factors surface and exercise direction evaluated movement symmetry changes (*p* < 0.05, Bonferroni correction). Withers movement indicated increased forelimb push-off with the re-shod limb on the inside of the circle and reduced weight-bearing with the re-shod limb and the ipsilateral hind limb on hard ground compared to soft ground. Movement symmetry measurements indicate that a rolled rocker shoe allows for increased push-off on soft ground in trot in a circle. Similar studies should study different types of shoes for improved practically relevant knowledge about shoeing mechanics, working towards evidence-based preventative shoeing.

## 1. Introduction

Horses undergo “routine” hoof care, trimming, and/or shoeing at regular intervals. In the context of preventative approaches, this puts hoof care providers in a unique position to gather quantitative evidence about changes in force production and movement prior to the occurrence of issues affecting the musculoskeletal system. For example, in Thoroughbred racehorses, detailed analysis of force platform data has shown potential in detecting subclinical tendon injuries at an early stage [1], and retrospective analysis of “in-race” stride parameters enables the detection of impending injuries up to six races prior to their occurrence [2].

With technological advances in equine gait analysis, easy-to-use tools such as inertial measurement units (IMUs) have led to an improved understanding of the associations between routinely used visual indicators of movement deficits—head nod [3] and hip hike [4]—and the ground reaction forces involved in the weight-bearing and push-off phases of the stride cycle in trotting horses [5,6]. They also allow for the quantification of more complex compensatory mechanisms [7], as well as back movement [8,9], which is relevant for the diagnosis and prevention of poor performance given the interaction between lameness and disorders of the thoracolumbosacral region [10].

Visits at regular intervals combined with practical, easy-to-use methods for quantifying functional aspects of locomotion allow for the gathering of large databases. It is essential that suitable protocols are established, minimizing variations (such as sensor or marker placement [11]) and taking into account normal variation of gait patterns [9,12]. Re-shoeing with open-heel shoes appears to leave fundamental ground reaction forces and their moment arms around the distal limb structures largely unchanged [13]. Considerable changes to ground reaction forces and kinematics, however, appear to occur over the course of the shoeing cycle [14] and in relation to more complex shoeing adaptations, such as for example wedges or toe modifications [15,16,17,18,19]. Thus, establishing normal ranges for the effects of different types of shoeing appears to be of practical importance.

In comparison to very detailed measurements of ground reaction forces with force platforms [1], the quantification of upper body movement symmetry provides less-detailed information. However, the functional aspect, i.e., the association of movement symmetry with the fundamentals of weight-bearing and push-off [5,6], combined with its ability to be easily measurable in the field, both on hard and soft ground [20,21] and during circular movement [22,23], emphasizes the practical relevance. 

Different shoe types are associated with complex interactions between breakover duration, the magnitude of the ground reaction force moment arm, and the magnitude of force [24]. A rolled rocker shoe is specifically designed to shorten the moment arm of the ground reaction force in late stance as well as the duration of breakover; however, it might increase the forces at the onset of breakover [24]. Shoeing with this type of shoe, as an example for an “orthopedic shoe” designed to alter specific aspects of force production, provides an opportunity to investigate typical ranges for the magnitude of kinematic gait changes in association with shoeing. 

The aim of this study was to provide quantitative data in a specific group of horses preventatively shod according to a specific shoeing regimen with rolled rocker shoes and to establish the magnitude of associated gait changes. We hypothesized that, analogously to the absence of changes in ground reaction force moment arms after shoeing with standard open heel shoes [13], the effects of rolled rocker shoeing on movement symmetry would be small and more related to push-off in the second half of stance. We specifically hypothesized that any effects related to the presumed shortening of the ground reaction force moment arm after trimming and re-shoeing would be more obvious on the soft surface, which allows for more efficient ground penetration of the re-shod foot during breakover. During circular exercise, changes might be more exacerbated for the inside limb, which deviates more from the vertical [25], gain allowing for a more effortless ground penetration with the re-shod foot.

## 2. Materials and Methods

### 2.1. Horses

A convenience sample of ten Warmblood horses actively participating in show jumping competitions of at least 1.20 m fence height were included in this study (see Table 1). Ages ranged from 7 to 13 years (median 11 years). There were seven geldings and three mares. All horses were deemed fit to compete by their owner prior to inclusion in the study. Horses were due for routine re-shoeing by their farrier (CB) as per their normal 4-week shoeing schedule. All horses included in this study were currently shod in rolled rocker shoes as part of their routine, preventative shoeing regimen, aiming to benefit from a shortening of the DIP moment arm analogous to previous findings, for example, with natural balance shoes [24]. Ethical approval was granted via the Royal Veterinary College Clinical Research Ethical Review Board (CRERB: URN 2019 1929-3). Written owner’s consent was obtained prior to inclusion.

### 2.2. Re-Shoeing

Rolled rocker shoes were applied according to the principles of a pre-defined hoof mapping system (https://en.shoeing4soundness.ch/, accessed on 22 July 2024) with the aim of fitting the shoe in relation to the approximate location of the presumed center of rotation of the hoof. The approximate center of rotation was found from external landmarks by first marking the widest part of the medial and lateral white line, drawing a line mediolaterally across the foot from those marks, and marking the point where the line crosses the center of the frog (see Figure 1). The aim was then to fit the shoe with equal proportions dorsal and palmar to that approximate location. During this process, measurements were conducted with a standard tape measure (see Figure 1).

### 2.3. Movement Symmetry Analysis

Upper body movement symmetry was analyzed with inertial measurement units (MTw, Xsens, Enschede, The Netherlands; ±2000 deg/s/±160 m/s^2^/±1.9 Gauss). The sensors were attached with doubled-sided tape to the head piece (over the poll), over the withers, and between the two tubera sacrale (i.e., over the sacrum). Three repeats of gait assessments were conducted:Prior to re-shoeing at the end of a 4-week shoeing cycle with the “old” shoes.After re-shoeing of a randomly selected front limb; i.e., after removal of the old shoe, trimming, and re-shoeing.After the re-shoeing of the second front limb.

For each gait assessment, measurements were taken under four different conditions, subjectively aiming at collecting at least 25 strides per condition:In-hand trot on the straight on hard ground (asphalt).Trot on well-maintained synthetic footing (Geo Textile—a combination of shredded textile fibers with sand):
In-hand on the straight;Lunged on a 10 m circle to the right;Lunged on a 10 m circle to the left.

Data were transmitted at a sample rate of 100 Hz per individual data channel (tri-axial acceleration, tri-axial rate of turn, tri-axial magnetic field) from the sensors mounted on the horse to a nearby laptop computer running MTManager version 4.8 (Xsens, Enschede, The Netherlands) software and connected to an Awinda transceiver station (Xsens, Enschede, The Netherlands). Following successful data collection, processing was performed with a combination of MTManager v. 4.8 (Xsens, Enschede, The Netherlands) and EquiGait v. 3w (Cheshunt, UK) software. 

Three vertical displacement asymmetry measures quantifying differences between minima (Dmin), between maxima (Dmax), and between upward amplitudes (Dup) between the two halves of each stride cycle were calculated. This resulted in nine symmetry parameters, three for the head (H), three for the withers (W), and three for the sacrum (pelvis: P) sensor: HDmin, HDmax, HDup, WDmin, WDmax, WDup, PDmin, PDmax, and PDup. In addition, the range of motion of vertical displacement was calculated for each of the three sensors: HROM, WROM, and PROM. Stride cycles were identified from continuous data streams using published stride segmentation methods [26]. Median values for each exercise condition (straight hard, straight soft, left circle soft, right circle soft) were tabulated and labeled according to the re-shoeing condition (“pre”: pre re-shoeing; “between”: after the trimming and re-shoeing of the first limb; “post”: after the trimming and re-shoeing of the second limb). Horse ID, surface (“hard” or “soft”), re-shod limb (“left” or “right”), and exercise condition (“straight”, “left circle”, “right circle”) were also tabulated together with stride time (in milliseconds) and the number of strides analyzed for each condition.

### 2.4. Study Design

Baseline gait analysis was conducted prior to trimming and re-shoeing at the end of a 4-week shoeing cycle.

Then, for each horse, the first forelimb undergoing re-shoeing (left or right) was randomly selected via coin toss. A rolled rocker shoe was applied to the selected foot according to the external reference points, and gait measurements were then repeated within approximately 10 min of the re-shoeing process in the same manner as during the baseline gait assessment. 

Finally, the second forelimb was re-shod to the same foot mapping system, and gait measurements were repeated. 

We use the term “limb-by-limb” re-shoeing for this specific re-shoeing protocol, which was implemented to benefit from the movement symmetry analysis that compares the movement parameters between the two halves of a trot stride cycle (see previous section: movement symmetry parameters), as well as to benefit from the implemented data normalization (see following section).

Shoes were applied via hot-shoeing with six copper E-head nails per shoe. The hind limbs were left “unchanged”, i.e., with the “old” shoes, in order to avoid compensatory movement changes influencing the movement symmetry results.

### 2.5. Data Normalization

Differences between “pre” and “post” trimming and re-shoeing values were calculated for movement asymmetry and range of motion values, resulting in twelve outcome parameters characterizing “pre/post” changes in movement between the periods before and after the application of the first shoe and between the periods before and after the application of the second shoe (nine movement asymmetry difference variables: DHDmin, DHDmax, DHDup, DWDmin, DWDmax, DWDup, DPDmin, DPDmax, DPDup; three range of motion difference variables: DHROM, DWROM, DPROM). Pre/post differences were calculated by subtracting the “pre” value from the “post” value. 

Since, numerically, it is likely that the effects of trimming and/or re-shoeing a left limb and trimming and/or re-shoeing of a right limb cause directionally opposite effects, all movement asymmetry differences after trimming/re-shoeing right limbs were inverted (multiplied by negative one). In addition, lunge direction was coded (tabulated) as “inside” and “outside” in relation to whether the re-shod limb was on the inside of the circle (i.e., left limb on left circle or right limb on right circle) or on the outside of the circle (i.e., left limb on right circle or right limb on left circle).

This data normalization procedure results in outcome variables (movement asymmetry differences) where increasing asymmetries, i.e., an indicator of reduced force production with the re-shod limb, are characterized by negative values, and increased asymmetries resulting from the contralateral limb are characterized by positive values. 

### 2.6. Statistical Analysis

#### Quantifying Differences between Left and Right Limb Shoeing Interventions

Statistical analysis was performed in SPSS software (version 29.0.0.0 (241), IBM, Chicago, IL, USA). Left versus right limb intervention differences were normally distributed (Lilliefors test) for all variables except DWDmax, DHROM, and DPROM. Accordingly, mean or median differences between left and right limb interventions were calculated (see Table 2). In order to evaluate whether the data of changes in movement pre/post trimming of left and right limbs could be “pooled” in the side-normalized dataset, and the side of re-shoeing consequently modeled as a “random effect”, movement asymmetry changes were compared to published 95% confidence intervals for test–retest repeatability. The 95% confidence intervals were chosen as 6 mm for head movement and 3 mm for pelvic movement [27]. Due to the absence of corresponding test–retest repeatability values for withers movement asymmetry, the more stringent value of 3 mm was applied (see Table 2, bottom row). 

None of the absolute values of the mean or median differences exceeded the pre-defined test–retest repeatability values (see Table 2). Consequently, the side of intervention (re-shod limb: left or right) was treated as a random variable for further statistical testing based on the side-normalized data (see description of mixed model).

Linear mixed models with “horse” and “limb within horse” as random factors and “surface” (hard or soft) and “direction” (straight-line, inside circle, outside circle) as fixed factors were implemented with pre/post differences for the nine movement asymmetry variables, as well as for the pre/post differences for the three range of motion variables as outcome variables. A Bonferroni correction was used for “direction”. The level of significance was set to *p* < 0.05. Estimated marginal means and their 95% confidence intervals were calculated for all fixed-factor categories to characterize typical changes before/after re-shoeing with rolled rocker shoes.

## 3. Results

### 3.1. Baseline Movement Symmetry before Trimming and Re-Shoeing

At baseline gait analysis (Table 3), the ten Warmblood show-jumping horses showed median movement symmetry values on the hard surface of between −9.5 mm (left asymmetry) for WDup and +7.0 mm (right asymmetry) for HDup, and range of motion values varied between 77.0 mm for head movement and 98.0 mm for pelvic movement (Table 3). On the soft surface, median movement symmetry varied between −8.0 mm (left asymmetry) for WDup and +2.5 mm (right asymmetry) for PDmax, and range of motion values varied between 80.5 mm for head movement and 102.0 mm for pelvic movement (Table 3). Refer to Table 3 for additional values (25th and 75th percentile).

During circular exercise at baseline gait analysis, the ten Warmblood show jumping horses showed median movement symmetry values (see Table 4 for 25th and 75th percentile) on the left rein of between −18.0 mm (left asymmetry) for WDmin and +6.5 mm (right asymmetry) for PDmax, and range of motion values between 86.0 mm for head movement and 106.5 mm for pelvic movement (Table 4). On the right rein, median movement symmetry varied between −8.5 mm (left asymmetry) for WDmax and +7.0 mm (right asymmetry) for PDup, and range of motion varied between 87.5 mm for head movement and 109.0 mm for pelvic movement (Table 4). 

### 3.2. Effect of Surface and Movement Direction on Changes after Routine Trimming and Shoeing with Rolled Rocker Shoes

Table 5 shows the results of mixed model analysis—level of significance, estimated marginal mean values, and 95% confidence intervals—assessing the effect of re-shoeing with rolled rocker shoes on movement asymmetry and range of motion. Each mixed model was based on the analysis of N = 80 movement symmetry values: 10 horses × 2 legs × 4 exercise conditions (hard straight; soft straight; soft inside rein; soft outside rein). DWDmax was the only movement asymmetry variable significantly affected by the direction of movement. A small positive estimated marginal mean value of +1.25 mm for trot with the re-shod limb on the inside of the circle indicates an increase in weight-bearing with the re-shod limb. The small negative estimated marginal mean values of −0.725 mm with the re-shod limb on the outside of the circle and a very small negative value of −0.225 mm for straight-line trot indicate minimally reduced weight bearing compared to the contralateral limb under these conditions.

Three movement symmetry variables—DHDup, DWDmin, and DPDmax—were significantly affected by surface (Table 5). DHDup, a head movement asymmetry variable comparing the two upward movement amplitudes, indicates decreased push-off with the re-shod limb compared to the contralateral limb on the hard surface (estimated marginal mean: −4.017 mm) and increased push-off on the soft surface (estimated marginal mean: +2.583 mm). Withers movement asymmetry, DWDmin, comparing the two most downward positions, shows a similar trend. A clear reduction in weight-bearing with the re-shod limb compared to the contralateral limb is observable on the hard surface, with an estimated marginal mean value of −2.9 mm. A small reduction of −0.6 mm is measurable on the soft surface. Pelvic movement, DPDmax, comparing the two most upward positions of the pelvis, showed evidence of reduced push-off with the hind limb ipsilateral to the re-shod forelimb. This is observed for both hard and soft ground, with estimated marginal mean values of −2.017 mm and −0.267 mm, respectively.

Range of motion differences between pre/post re-shoeing were affected neither by the direction of exercise nor by the surface on which the exercise was performed. 

## 4. Discussion

### 4.1. Effect of Surface and Movement Direction on Changes after Trimming and Re-Shoeing with Rolled Rocker Shoes

In order to provide practically and functionally relevant evidence, i.e., data from exercises that are relevant for the day-to-day activities of horses, each horse was assessed on two different surfaces, as well as during straight-line trot and during circular movement on soft ground and with validated inertial measurement unit technology [28]. While the pre/post re-shoeing-related movement differences of all three investigated upper body landmarks (head, withers, and pelvis) were affected by the type of surface that the horses were exercised on, only one parameter—associated with withers movement—was affected by the direction of movement, emphasizing the importance of assessing shoeing-related effects on different surfaces. 

The positive estimated marginal mean value of approximately +2.6 mm for DHDup on the soft surface (Table 5)—a parameter associated with the upward movement amplitudes of the head and, consequently, with forelimb push-off—indicates that the newly fitted rolled rocker shoe affords the horse the ability to push off more efficiently on soft ground. We speculate that this might be related to the process of easing the breakover process and allowing the toe region of the hoof to penetrate the ground in a manner that is more efficient for pushing off. However, a shortening of the breakover duration, which may indicate a more effortless breakover process, in conjunction with a similar shoe (with a rolled toe), has only been reported in walk [29]. Our IMU-based approach does not directly measure force, so ultimately, further studies might be required in order to understand the exact mechanism. Force plate analysis is difficult but not impossible on soft ground [30].

The negative estimated marginal mean value of approximately −4 mm on hard ground, on the other hand, indicates that the rolled rocker shoe is not supportive of push-off in the same manner on hard ground. It is associated with a reduced push-off effort with the re-shod limb. Again, to elucidate the underlying mechanism(s), both force and movement would need to be quantified simultaneously. The focus of our present study is a functional assessment method, which can be implemented in both clinical and farriery practice. Here, specifically, we are interested in the magnitude of changes (estimated marginal mean values and 95% confidence intervals) that can be expected after “routine” re-shoeing with a shoe that is “familiar” to the horse. Future studies could use this as a reference when studying the effect of shoes that are more unfamiliar to a horse, for example, in relation to shape or material. It should, however, be emphasized that the maximum benefit of using symmetry-based assessments would require a deviation from standard farriery practice and ideally implement a “limb-by-limb” shoeing protocol, as used in the present study.

With respect to movement direction, there appears to be a benefit to the horses from the rolled rocker shoe, particularly when the newly fitted shoe is on the inside of the circle. Under that condition, the positive estimated marginal mean of +1.25 mm for withers movement (DWDmax) indicates increased push-off according to our data normalization procedure. It is feasible that the modified toe shape (possibly together with the effect of trimming and shortening the base of support) allows for more efficient ground penetration. This might allow for improved mediolateral alignment of the hoof with the distal limb, particularly for the inside limb, which shows an increased deviation from the vertical in trot on a curve [25]. However, the amount of rotation of the hoof has not been quantified in the present study, and further investigation is needed to support this speculation. Hoof orientation during straight-line walking, for example, has shown limited changes in mid stance, even with more “extreme” shoe modifications such as wedges or bar shows [31]. 

### 4.2. The “Limb-by-Limb” Data Collection Protocol

The shoeing regimen was left unchanged in all horses of this study; they had all been shod previously with the same rolled rocker shoe and the same fitting regimen according to a specific hoof mapping system; hence, we use the term “re-shoeing” here. The detailed re-shoeing process was, however, modified from standard procedures, generally trimming and (re-)shoeing pairs of limbs. Here, one front limb was chosen at random, and the full re-shoeing process performed for that limb, i.e., the old shoes were removal and the hoof was trimmed and then re-shod. Then, the same process was repeated for the contralateral limb. This allowed us to study the effects of re-shoeing for each limb separately. While this process has allowed us to study the effects of re-shoeing, it has not allowed us to differentiate between the effects of trimming from those of shoeing [32]. The specific trimming and shoeing process used here included a shortening of the dorsal hoof wall and positioning of the shoe in relation to a specific hoof mapping protocol. As a median value across all reshod limbs, the distance between the dorsopalmar center of the shoe and the center of articulation of the distal interphalangeal joint (determined via hoof mapping) was shortened by 8 mm. It might be interesting to investigate further how shoe placement (and potentially the difference in placement between contra and lateral limbs) influences force distribution, and hence movement symmetry.

Together with the employed data normalization procedure, this “limb-by-limb” procedure allowed us to calculate the gait differences between the baseline assessment and after re-shoeing of the first limb, as well as a second difference between after re-shoeing of the first limb and after re-shoeing of the second limb. While this alters the flow of routine re-shoeing, it allows for a “consummate” quantification of the combined effect of the altered limb–ground interface together with the dynamic response of the horse. This quantification is achieved in the current study by utilizing the fundamental link between force asymmetry and movement asymmetry, which applies to the vertical movement of both the head and the pelvis [5,6].

Changes in how each horse uses pairs of contralateral limbs for exerting forces for weight-bearing and push-off can be undertaken through the quantification of changes in head and pelvic movement symmetry through previously established associations with force platform data [5,6]. In addition, withers movement provides information about compensatory movements that are not readily available through head and pelvic movement [7]. Had both forelimbs been re-shod “in one go” as opposed to our “limb-by-limb” approach, small movement changes in one direction, for example, as a response to the re-shoeing of the first limb, might have been counteracted by changes in the opposite direction after the re-shoeing of the contralateral limb. Overall, this might have resulted in near-zero asymmetry changes between the baseline assessment and the assessment after re-shoeing both limbs of a pair as performed in previous studies [33,34], and important information about the re-shoeing process would have been lost. It has to be emphasized that a “limb-by-limb” shoeing (or re-shoeing) process is preferable in order to benefit fully from movement symmetry analysis. 

### 4.3. Towards Guideline Values for Normal Changes

In addition to the average changes for our “limb-by-limb” re-shoeing approach, we have calculated 95% confidence intervals for estimated marginal mean values based on a mixed-model approach. This provides the first guideline values for the magnitude of change in upper body movement symmetry. Here, this assessment is restricted to one specific re-shoeing approach in one specific group of horses. It would appear advantageous to initiate the collection of a larger database for specific groups of horses and a variety of re-shoeing and exercise regimens. Ultimately, this would result in more robust guideline values for the expected gait changes after re-shoeing and might make it easier to detect injury-related gait changes earlier if, for example, larger changes are measured. 

Previous studies utilizing upper body movement symmetry measurements—including ours—have concentrated on the stages of the shoeing process [32] or have focused on changes over a longer period of 12 weeks and a transition from barefoot to open heel shoes [34]. Others have investigated the association between changes in hoof shape and gait symmetry [33]. However, to the authors’ knowledge, the present study is the first to have employed the “limb-by-limb” shoeing approach in combination with utilizing functional assessment of upper body movement symmetry pre/post the shoeing process. 

### 4.4. Study Population—Baseline Movement Symmetry

The horses included in this study consisted of a comparatively small convenience sample of ten horses under the routine care of one qualified farrier (CB) and were horses competing in show jumping. All horses had been routinely shod for a minimum of six months with rolled rocker shoes at the time of inclusion in the study; i.e., the horses were “re-shod” with the same type of shoe. This study was not designed to provide any indication about the use of rolled rocker shoes in horses not previously shod with these shoes or to analyze the effect of other horse-specific variables such as age, sex, height, or weight. It was most important in the context of the current study that the horses had been previously shod with the same type of shoe; i.e., they had been “re-shod”.

The horses were all considered “fit to compete” by their owners. At baseline gait analysis, absolute values of the median movement symmetry parameters across all horses for straight-line trot were ≤7 mm for head movement, ≤9.5 mm for withers movement, and ≤5 mm for pelvic movement (Table 3 and Table 4). The median values are thus smaller than the guideline values of ≤9 mm for head movement and ≤5 mm for pelvic movement, adapted from the published thresholds of 6 mm for head movement and 3 mm for pelvic movement [35,36] utilized during lameness examinations for identifying the most likely affected limb. However, interquartile ranges (Table 3) clearly exceed the guideline values, indicating that some movement asymmetry values in some of the study horses were outside the guideline values. Published withers movement symmetry values for non-lame horses are less commonly provided in the literature. A recent study indicates a “normal range” spanning from −10% to +7%, expressed as a percentage of the range of motion [37]. Using the average value of withers range of motion across hard and soft ground for our horses (99 mm), the “normal range” translates into −10 mm to +7 mm. This is similar to the median values shown here, and again, some horses exceed these values (Table 3). 

During circular trot on the lunge, movement symmetry changed in accordance with previously observed patterns [22,23,38] (compare Table 3 and Table 4). In general, with the exception of WDmax, increased left-sided asymmetry was measured on the left rein, and increased right-sided asymmetry was measured on the right rein. Absolute median values for circular trot are ≤7 mm for head movement, ≤18 mm for withers movement, and ≤10.5 mm for pelvic movement, similar to previously recorded values on hard and soft surfaces [22,38,39,40].

## 5. Conclusions

The “limb-by-limb” re-shoeing protocol, in combination with IMU-based measurements of upper body landmarks, allowed for a quantification of functional gait adaptations after re-shoeing with a specific shoe type under practically relevant exercise conditions. This approach could be useful for further studies with different re-shoeing regimens.

Increased forelimb push-off was found with the re-shod limb on the inside of the circle on soft ground, an effect that had not been apparent during straight-line assessment on hard ground. When riders are reporting differences after re-shoeing in relation to specific exercises, it may be advisable to perform a tailored gait assessment, encompassing both straight-line and circular trot, as well as different ground surfaces, since in the present study, hard and soft ground and circular movement have been related to significant re-shoeing-related movement changes across all three body landmarks. 

## Figures and Tables

**Figure 1 sensors-24-04848-f001:**
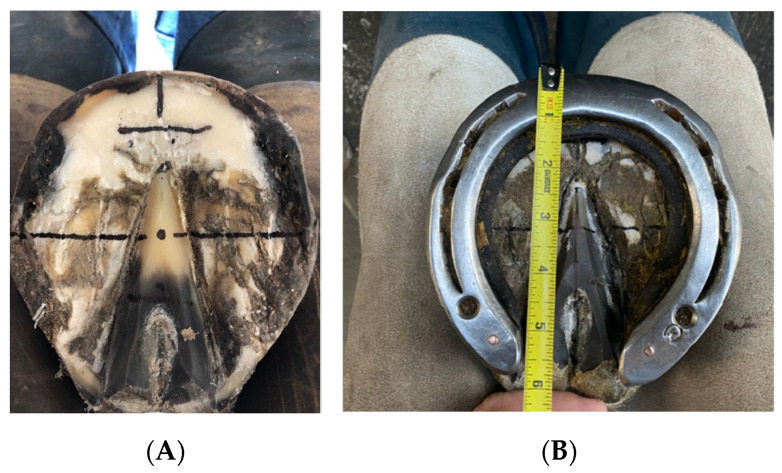
(**A**) External reference points marked with a pre-defined hoof mapping system used for fitting the rolled rocker shoe with reference to the presumed center of rotation. (**B**) Measurement of proportions of the shoe dorsal and palmar to the approximate center of rotation for fitting 50% of the shoe dorsal and palmar to the identified point. Measurements were undertaken with a tape measure.

**Table 1 sensors-24-04848-t001:** Age, sex, breed, competition level, and time in rolled rocker shoes prior to inclusion in the study.

Horse ID	Sex	Breed	Age (Years)	Time in Rolled Rocker Shoes	Competition Level
1000	Gelding	Dutch Warmblood	8	>12 shoeing cycles	>1.40 m
1001	Gelding	Dutch Warmblood	13	>12 shoeing cycles	>1.45 m
1002	Mare	Dutch Warmblood	7	>12 shoeing cycles	>1.40 m
1003	Gelding	Dutch Warmblood	11	>12 shoeing cycles	>1.30 m
1005	Gelding	Dutch Warmblood	11	>12 shoeing cycles	>1.30 m
1007	Gelding	Dutch Warmblood	11	6th shoeing cycle	>1.50 m
1008	Gelding	Dutch Warmblood	12	6th shoeing cycle	>1.30 m
1009	Mare	Dutch Warmblood	8	>12 shoeing cycles	>1.20 m
1010	Gelding	Dutch Warmblood	11	>12 shoeing cycles	>1.20 m
1011	Mare	Dutch Warmblood	10	>12 shoeing cycles	>1.20 m

**Table 2 sensors-24-04848-t002:** Mean (and median) differences (D) between left and right limb interventions. H: head; W: withers; P: pelvis; Dmin: difference between minima; Dmax: difference between maxima; Dup: difference between upward movement amplitudes; ROM: range of motion, i.e., difference between highest maximum and lowest minimum.

	DHDmin (mm)	DHDmax (mm)	DHDup (mm)	DHROM (mm)	DWDmin (mm)	DWDmax (mm)	DWDup (mm)	DWROM (mm)	DPDmin (mm)	DPDmax (mm)	DPDup (mm)	DPROM (mm)
Median diff	2.5	−2.5	−1.0	1.0	−0.5	1.0	0.5	2.0	0.0	1.5	1	−1
Mean diff.	3.9	−0.35	4.7	NA	−0.6	−0.8	2.65	NA	−0.1	1.55	2.4	NA
Threshold	abs diff ≤ 6				abs diff ≤ 3				abs diff ≤ 3			

**Table 3 sensors-24-04848-t003:** Baseline movement symmetry during straight-line, in-hand trot on hard ground. Acronyms: see Table 2 for movement symmetry parameters; 25th per.: value of 25th percentile; 75th per.: value of 75th percentile; min: minimum value; max: maximum value. hard: measurements on hard ground; soft: measurements on soft ground.

	Hard					Soft				
	Median	25th per.	75th per.	Min	Max	Median	25th per.	75th per.	Min	Max
HDmin (mm)	3.0	−4.0	14.5	−18	+26	−0.5	−11.0	10.75	−21	27
HDmax (mm)	1.0	−3	5.5	−11	9	−3.0	−8.75	2.5	−22	6
HDup (mm)	7.0	−7.25	16	−22	38	−3.5	−18.0	6.0	−44	27
HROM (mm)	77.0	70.75	79.75	58	93	80.5	73.75	88.0	62	103
WDmin (mm)	−4.0	−7.5	−1.0	−11	4	−4.5	−7.0	−0.5	−11	8
WDmax (mm)	−5.0	−9.0	4.0	−12	6	−4.5	−8.0	0.5	−14	5
WDup (mm)	−9.5	−13.0	1.75	−22	11	−8.0	−14.5	−0.5	−24	9
WROM (mm)	96.5	91.0	105.0	84	108	101.5	94.0	112.25	87	117
PDmin (mm)	−3.5	−6.25	3.5	−7	13	−5.0	−5.25	0.25	−6	12
PDmax (mm)	2.0	−4.0	7.5	−8	24	2.5	−4.0	7.25	−12	19
PDup (mm)	−2.5	−7.25	7.25	−12	39	−2.0	−7.5	2.75	−13	28
PROM (mm)	98.0	91.5	102.0	76	112	102.0	98.75	111.0	76	112

**Table 4 sensors-24-04848-t004:** Baseline movement symmetry during trot on the lunge on the left and right circles. Left circle: measurements from lunging in trot on the left rein; right circle: measurements from lunging in trot on the right rein. All other acronyms: see Table 2 and Table 3.

	Left Circle					Right Circle				
	Median	25th per.	75th per.	Min	Max	Median	25th per.	75th per.	Min	Max
HDmin (mm)	−2.0	−11.75	11.25	−35	20	5.0	−6.0	16.5	−21	23
HDmax (mm)	−7.0	−12.75	0.5	−16	15	1.0	−6.25	4.25	−21	8
HDup (mm)	−4.0	−22.25	7.5	−50	28	3.5	−6.5	15.75	−36	32
HROM (mm)	86.0	81.0	91.25	73	93	87.5	78.5	91.5	76	94
WDmin (mm)	−18.0	−21.5	−11.5	−29	−8	8.0	2.0	15.0	2	18
WDmax (mm)	0.5	−3.0	6.25	−7	13	−8.5	−12.25	−3.5	−14	1
WDup (mm)	−15.5	−24.5	−7.75	−32	1	3.0	−9.25	9.5	−13	17
WROM (mm)	103.0	97.5	112.75	87	122	102.5	92.25	113.75	90	122
PDmin (mm)	−10.5	−21.25	−3.75	−23	0	5.0	3.0	18.5	0	21
PDmax (mm)	6.5	−0.75	12.25	−10	16	1.0	−3.0	6.5	−11	17
PDup (mm)	−7.0	−15.0	1.0	−18	8	7.0	3.5	22.75	−9	36
PROM (mm)	106.5	100.0	113.75	96	123	109.0	102.75	115.25	89	118

**Table 5 sensors-24-04848-t005:** *p*-values and estimated marginal mean (EMM) values (with 95% confidence intervals) of mixed-model analysis for pre/post trimming and re-shoeing differences for head, withers, and pelvic movement asymmetry, and range of motion variables for movement direction (inside rein, outside rein, straight line) and for hard and soft surfaces. Significant *p*-values (<0.05) are shown in bold. Estimated marginal mean values and confidence intervals (conf. intv.) are given in millimeters. Movement variable acronyms: see Table 2. Increasing asymmetries, i.e., indicators of reduced force production with the re-shod limb, are characterized by negative values, and increased asymmetries resulting from the contralateral limb are characterized by positive values.

	*p*-Value		EMM (conf. intv.) Direction			EMM (conf. intv.) Surface	
Parameter	Direction	Surface	Inside (L)	Outside (R)	Straight	Hard	Soft
DHDmin (mm)	0.946	0.068	−0.800 (−4.401, 2.801)	−0.450 (−4.051, 3.151)	−0.150 (−2.486, 2.186)	−2.267 (−6.038, 1.504)	1.333 (−0.717, 3.384)
DHDmax (mm)	0.411	0.070	0.425 (−2.670, 3.520)	−1.275 (−4.370, 1.820)	0.725 (−1.385, 2.835)	−1.517 (−4.746, 1.713)	1.433 (−0.464, 3.331)
DHDup (mm)	0.319	**0.038**	−0.800 (−6.620, 5.020)	−3.050 (−8.870, 2.770)	1.700 (−2.114, 5.514)	−4.017 (−10.107, 2.074)	2.583 (−0.782, 5.948)
DHROM (mm)	0.721	0.795	0.775 (−2.364, 3.914)	0.975 (−2.164, 4.114)	−0.325 (−2.308, 1.658)	0.250 (−3.043, 3.543)	0.700 (−1.016, 2.416)

DWDmin (mm)	0.313	**0.017**	−2.550 (−4.313, −0.787)	−1.550 (−3.313, 0.213)	−1.150 (−2.362, 0.062)	−2.900 (−4.742, −1.058)	−0.600 (−1.715, 0.515)
DWDmax (mm)	**0.016**	0.107	1.250 (0.012, 2.538)	−0.725 (−1.988, 0.538)	−0.225 (−1.005, 0.555)	0.683 (−0.643, 2.010)	−0.467 (−1.132, 0.199)
DWDup (mm)	0.639	0.439	−1.200 (−3.344, 0.944)	−2.250 (−4.394, −0.106)	−1.450 (−2.892, −0.008)	−2.083 (−4.327, 0.160)	−1.183 (−2.499, 0.132)
DWROM (mm)	0.215	0.649	0.325 (−2.127, 2.777)	−1.775 (−4.227, 0.677)	−1.925 (−3.341, −0.509)	−1.450 (−4.035, 1.135)	−0.800 (−1.956, 0.356)

DPDmin (mm)	0.895	0.859	1.000 (−0.941, 2.941)	0.600 (−1.341, 2.541)	1.100 (−0.020, 2.220)	0.800 (−1.246, 2.846)	1.000 (0.085, 1.915)
DPDmax (mm)	0.529	**0.033**	−1.325 (−2.884, 0.234)	−0.625 (−2.184, 0.934)	−1.475 (−2.596, −0.354)	−2.017 (−3.641, −0.393)	−0.267 (−1.315, 0.781)
DPDup (mm)	0.999	0.230	−0.350 (−3.056, 2.356)	−0.400 (−3.106, 2.306)	−0.400 (−1.962, 1.162)	−1.333 (−4.185, 1.519)	0.567 (−0.709, 1.842)
DPROM (mm)	0.091	0.869	−0.100 (−2.293, 2.093)	1.800 (−0.393, 3.993)	−0.800 (−2.235, 0.635)	0.400 (−1.899, 2.699)	0.200 (−1.094, 1.494)

## Data Availability

The data analyzed in the present manuscript are available via the following FigShare link 10.6084/m9.figshare.26364625.

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
