# Peer review of "Inertial Sensor-Based Quantification of Movement Symmetry in Trotting Warmblood Show-Jumping Horses after “Limb-by-Limb” Re-Shoeing of Forelimbs with Rolled Rocker Shoes"

_sensors, 2024, doi:10.3390/s24154848_

Round 1

Reviewer 1 Report

Comments and Suggestions for Authors

Overall a very well written paper with good experimental design and execution.  There are a few areas where further explanation would be beneficial.

Line 110 - Xsens makes a number of sensors please provide model of sensor.  Specifics about the sensor, such as ranges or limits may also be relevant to readers.  

Lines 165-167 - Please reword this sentence it is quite awkward and does not convey what you are trying to explain well.

Statistical analysis -Was a statistics software used to complete the statistics such as SAS or R?  if so please cite the software and version (if applicable).

All Tables: Add units into the table proper next to the variable name example: HDmin (mm).  Then the "All values in millimeters" can be removed from the table description.

Table 4: Update the table description to include what data the table is presenting.  pre/post shoeing differences?  At the present time it is unclear especially without referencing line 196-202. A one sentence description of the data would be sufficient to allow the table to stand alone. 

Discussion - 

The authors discuss why trimming and reshoeing happened 1 hoof at a time and why that was important to the design, but they do not discuss how hoof length differences may have impacted results.  How much hoof was removed during trimming and was it similar between hooves and horses?  How might the difference in hoof length effect results?  

Author Response

Comment 1:Overall a very well written paper with good experimental design and execution.  There are a few areas where further explanation would be beneficial.

Response 1: Thank you for your considerate review of our manuscript.

Comment 2: Line 110 - Xsens makes a number of sensors please provide model of sensor.  Specifics about the sensor, such as ranges or limits may also be relevant to readers.  

Respone 2: Thank you for highlighting this omission. We have added the type of sensor as well as the ranges for the accelerometers, gyroscopes and magnetometers. The sample rate used (per individual data channel) is mentioned a few sentences later, where the data collection is described. We hope it makes sense to separate the fixed ‘physical properties’ such as the measurement ranges from the description of the ‘adjustable properties’ such as the sample rate, which can be adjusted for different needs of data collection.

Comment 3: Lines 165-167 - Please reword this sentence it is quite awkward and does not convey what you are trying to explain well.

Response 3: We have rephrased this sentence: Since numerically it is likely that effects of trimming and/or shoeing of a left limb and trimming and/or shoeing of a right limb cause directionally opposite effects, all movement asymmetry differences after trimming/shoeing right limbs were inverted (multiplied by negative one).

Comment 4: Statistical analysis -Was a statistics software used to complete the statistics such as SAS or R?  if so please cite the software and version (if applicable).

Response 4: Thank you again for highlighting a clear omission. We have added that SPSS software has been used.

Comment 5: All Tables: Add units into the table proper next to the variable name example: HDmin (mm).  Then the "All values in millimeters" can be removed from the table description.

Response 5:  Thank you for this suggestion. We have amended the tables as requested.

Comment 6: Table 4: Update the table description to include what data the table is presenting.  pre/post shoeing differences?  At the present time it is unclear especially without referencing line 196-202. A one sentence description of the data would be sufficient to allow the table to stand alone. 

Response 6: Thank you. We have clarified that this table presents pre/post trimming and re-shoeing differences. We have also added a sentence to the table clarifying the ‘meaning’ of positive and negative values to save the reader from going back to the main text to find this information.

Comment 7: Discussion - The authors discuss why trimming and reshoeing happened 1 hoof at a time and why that was important to the design, but they do not discuss how hoof length differences may have impacted results.  How much hoof was removed during trimming and was it similar between hooves and horses?  How might the difference in hoof length effect results?  

Response 7: Thank you for this comment. We have some additional data about the trimming and shoeing process and have added a couple of sentences to the discussion on this topic.
In reference to the limb-by-limb protocol, we have added the following sentences: “While this process has allowed us to study the effects of re-shoeing, it has not allowed us to differentiate between the effects of trimming and the effects of shoeing [32]. The specific trimming and shoeing process used here included a shortening of the dorsal hoof wall and positioning of the shoe in relation to a specific hoof mapping protocol. As a median value across all-reshod limbs, the distance between the dorso-palmar center of the shoe and the center of articulation of the distal interphalangeal joint (determind via hoof mapping) was shortened by 8 mm. It might be interesting to further investigate how shoe placement (and the difference in placement between contra-lateral limbs) influences force distribution and hence movement symmetry."

Reviewer 2 Report

Comments and Suggestions for Authors

The farrier industry is a large aspect of the horse industry, and the importance of farrier work on the sport horse has been well documented. With the introduction of accessible, new technologies such as that of the inertial sensors utilized in the current study, further work looking at specific shoes and shoeing methodology can be carried out. As such, this study can have a practical application to the horse industry and specifically to the farrier industry.

While this study has the potential to hold value, it is unclear from the title and throughout the manuscript as to what is being tested. At times, sections can be misleading. From the title and abstract, it would be assumed that these horses were being reshod for the first time with a rolled rocker shoe, but it is later the reader learns that the horses are just being reshod with what they are already familiar with. There is no mention, even through the introduction, of the 'limb by limb' approached being assessed. However, this is the main area where significant differences are found. Could these findings be related to the fact that the horse was unbalanced by the longer hoof wall that had yet to be trimmed and not the reshoeing using a familiar shoe? How often do farriers just trim and shoe one hoof? As quadrupeds, the only way to isolate the mechanics of a singular limb is through a cadaver study. Further, while the changes observed with the one forelimb was attributed solely to the fact that the forelimb had been reshod unlike the other forelimb, authors make no mention within the methodology as to what was done with the hindlimbs and the timing associated with the farrier work of the hindlimbs compared to that of the forelimbs being studied. Was this timing consistent throughout? In the end, since the authors were not testing a new shoe, are they studying the value of shoeing both forelimbs at the same time, although trimming/shoeing one is not a common practice, were they testing the value of trimming/shoeing at four weeks, or something else? The impact of the shoe can't be isolated as the trimming itself could have caused some impact. The asymmetry observed with just one forelimb trimmed is obvious, and thus, why farriers do not treat a hoof independently unless there is a clinical application. Further, the asymmetry can also not be ruled solely to the trimming/shoeing of just one limb as it may be related more to the limb itself as without a thorough workup for all limbs for all horses utilizing diagnostic imaging, could the 'random' selection of a limb using 'coin toss' have led to a limb that already had potential lameness issues. The authors mention within the discussion that asymmetry was observed prior to the farrier work. It is not uncommon for the age range of the horses utilized within the show jumping industry to have functional unsoundness issues that would not prevent them from the sport. Without further details concerning the horses this conclusion has to be considered. A larger sample size would help. Further, follow up measurement in the next four-week reshoeing period looking at the other forelimb would assist in further investigating this conclusion. 

It is only within the discussion, lines 292-294 and lines 316-317, is it somewhat clear what the aims of the study are. Until then, it seems the focus is on the impact of the rolled rocker shoes. As such, starting with the title it needs to be clear what the aims are and the justification for the methods employed. It needs to be clear that only forelimbs were measured, although information about the hindlimb trimming and shoeing including timing of this work needs to be added to the methodology. Due to the small sample size it needs to be clear starting with the title that the study centers around Warmblood show jumpers as breed and performance type have been noted as influencing changes in biomechanics. It needs to be clear that these horses are getting reshod with the same shoe type they are familiar with. Was there any adjustment period before measurements as if not this can be justified by their familiarity with the shoe? Even the fact that this was a four-week reshoeing cycle as changes may be more significant with a longer period between. The limb by limb analysis should be removed as it holds no relevance to the industry as that is not a common practice on shoeing one limb. The asymmetry observed is why farriers trim/shoe both limbs. Further details need to be provided for all horses including clinical assessments and histories. A control needs to be included looking at just trimming itself with the horses unshod to rule out the potential influence of changes in the angles due to trimming compared to that of the use of the rolled rocker shoe. Without these changes, the contribution of this manuscript more lies in the lines 337-352 concerning the IMU based measurements of the upper body landmarks. This information can be of value to the industry if the manuscript is revised to a methodology paper in which the small sample size and lack of control would be less of an issue if this manuscript more focused on the methodology and avoided making any conclusions concerning the shoeing. 

Author Response

Comment 1: The farrier industry is a large aspect of the horse industry, and the importance of farrier work on the sport horse has been well documented. With the introduction of accessible, new technologies such as that of the inertial sensors utilized in the current study, further work looking at specific shoes and shoeing methodology can be carried out. As such, this study can have a practical application to the horse industry and specifically to the farrier industry.

Response 1: Thank you for your general assessment.

Comment 2: While this study has the potential to hold value, it is unclear from the title and throughout the manuscript as to what is being tested. At times, sections can be misleading. From the title and abstract, it would be assumed that these horses were being reshod for the first time with a rolled rocker shoe, but it is later the reader learns that the horses are just being reshod with what they are already familiar with.

Response 2:Thank you for indicating that this has been unclear to you. We are not sure how else to make it clear that the horses had been shod with the same type of shoes as previously. We are specifically using the term “re-shoeing” (or “re-shod”) throughout the manuscript and in the title and have also added the term “limb-by-limb” to the title of the manuscript to highlight this specific aspect of the study. Both terms “re-shoeing” and “limb-by-limb” had already been introduced in the original version of the abstract:
“Ten Warmblood show jumping horses (7-13 years; 7 geldings, 3 mares) underwent forelimb re-shoeing with rolled rocker shoes, one limb at a time (“limb-by-limb”).

We have also added more information about the “limb-by-limb” protocol and the associated term in the new version of the materials and methods section:
“We use the term “limb-by-limb” re-shoeing for this specific re-shoeing protocol that was implemented to benefit from the movement symmetry analysis that compares the movement parameters between the two halves of a trot stride cycle (see previous section movement symmetry parameters) as well as to benefit from the implemented data normalization (see following section).”

We have carefully checked the manuscript and are now much more consistently using the term “re-shoeing” when referring to the procedures and effects of the current study. Where we are referring to ‘general principles’ we are still using the term “shoeing”.

Comment 3: There is no mention, even through the introduction, of the 'limb by limb' approached being assessed.

Response 3:We have added more information to the materials and methods section and are now more explicitly introducing the term ‘limb-by-limb’ and are also more consistent using the term ‘re-shoeing’. It is difficult to introduce the term ‘limb-by-limb’ in the introduction, when there is no relevant literature available for this term.

Comment 4: However, this is the main area where significant differences are found. Could these findings be related to the fact that the horse was unbalanced by the longer hoof wall that had yet to be trimmed and not the reshoeing using a familiar shoe? How often do farriers just trim and shoe one hoof? As quadrupeds, the only way to isolate the mechanics of a singular limb is through a cadaver study.

Response 4: We respectfully disagree and we discuss this in the discussion. We are using general mechanical principles relating movement symmetry with force asymmetry and by changing ‘foot mechanics’ one limb at a time we are actually assessing the dynamic reaction of the horse, i.e. how the horse is distributing force in association with ‘asymmetrical’ mechanical changes. It is impossible to use cadaver studies for assessing how the whole horse is dealing with these re-shoeing change.  

Comment 5: Further, while the changes observed with the one forelimb was attributed solely to the fact that the forelimb had been reshod unlike the other forelimb, authors make no mention within the methodology as to what was done with the hindlimbs and the timing associated with the farrier work of the hindlimbs compared to that of the forelimbs being studied. Was this timing consistent throughout?

Response 5: Thank you for indicating this. The materials and methods section mentions that re-shoeing was conducted at the end of a 4-week shoeing interval. The hind limbs were left ‘untouched’ not to introduce the possibility of compensatory movement patterns to influence the results. We have added a sentence to the materials and methods section explicitly mentioning this.

Comment 6: In the end, since the authors were not testing a new shoe, are they studying the value of shoeing both forelimbs at the same time, although trimming/shoeing one is not a common practice, were they testing the value of trimming/shoeing at four weeks, or something else?

Response 6: Thank you for your query. We have added more detail to the materials and methods section and the discussion is also addressing this point: the limb-by-limb protocol together with the data normalization are enabling us to study the ‘functional effects’ of this shoe, i.e. how the horse as a whole is reacting to this shoe. This cannot be done in a cadaver study which will only allow to study the ‘passive’ effects.

Comment 7: The impact of the shoe can't be isolated as the trimming itself could have caused some impact. The asymmetry observed with just one forelimb trimmed is obvious, and thus, why farriers do not treat a hoof independently unless there is a clinical application. Further, the asymmetry can also not be ruled solely to the trimming/shoeing of just one limb as it may be related more to the limb itself as without a thorough workup for all limbs for all horses utilizing diagnostic imaging, could the 'random' selection of a limb using 'coin toss' have led to a limb that already had potential lameness issues.

Response 7: Both limbs are being assessed with our limb-by-limb method. We are presenting comprehensive movement symmetry data (data often used for quantifying lameness in horses) from both straight line and lunge data, which provides quantitative data, rather than subjective opinion about the ‘lameness state’ of the horses, and thus contributes to our study being able to be compared to future studies with respect to the ‘quantitative asymmetry state’ of the horses.

Comment 8: The authors mention within the discussion that asymmetry was observed prior to the farrier work. It is not uncommon for the age range of the horses utilized within the show jumping industry to have functional unsoundness issues that would not prevent them from the sport. Without further details concerning the horses this conclusion has to be considered. A larger sample size would help. Further, follow up measurement in the next four-week reshoeing period looking at the other forelimb would assist in further investigating this conclusion. 

Response 8: both forelimbs have been assessed with the ‘limb-by-limb’ protocol.

Comment 9: It is only within the discussion, lines 292-294 and lines 316-317, is it somewhat clear what the aims of the study are. Until then, it seems the focus is on the impact of the rolled rocker shoes. As such, starting with the title it needs to be clear what the aims are and the justification for the methods employed.

Response 9: We have added the word ‘limb-by-limb’ to the title and have more consistently used the term ‘re-shoeing’ throughout the manuscript wherever we are referring to our study. The material and methods section introduces the term ‘limb-by-limb’ in the new version of the manuscript.

Comment 10: It needs to be clear that only forelimbs were measured, although information about the hindlimb trimming and shoeing including timing of this work needs to be added to the methodology.

Response 10: Thank you: we have added the word ‘forelimb’ to the title. This was a clear oversight at our end. Apologies. Information about the hind limbs has been added.

Comment 11: Due to the small sample size it needs to be clear starting with the title that the study centers around Warmblood show jumpers as breed and performance type have been noted as influencing changes in biomechanics. It needs to be clear that these horses are getting reshod with the same shoe type they are familiar with.

Response 11: We have used the term re-shoeing much more consistently in the new version of the manuscript.

Comment 12: Was there any adjustment period before measurements as if not this can be justified by their familiarity with the shoe?

Response 12: Thank you. We have added information about the time frame of the gait assessment conducted after re-shoeing.

Comment 13: Even the fact that this was a four-week reshoeing cycle as changes may be more significant with a longer period between.

Response 13: Previous studies with different lengths of the shoeing cycled are mentioned in the discussion, as are for example studies looking at transitioning from barefoot to shod conditions.

Comment 14: The limb by limb analysis should be removed as it holds no relevance to the industry as that is not a common practice on shoeing one limb.

Response 14: We respectfully disagree and have further clarified that this is the best way of utilizing modern, validated  movement symmetry based assessments associated with the fundamental mechanics of movement in the context of farriery.

Comment 15: The asymmetry observed is why farriers trim/shoe both limbs.

Response 15: We have addressed this before. The limb-by-limb process allows us to study the ‘consummate’ effects, i.e. how the whole horse deals with changed mechanics in a dynamic manner.

Comment 16: Further details need to be provided for all horses including clinical assessments and histories.

Response 16: Please see previous comment. We are providing comprehensive quantitative movement symmetry data for the horses.

Comment 17: A control needs to be included looking at just trimming itself with the horses unshod to rule out the potential influence of changes in the angles due to trimming compared to that of the use of the rolled rocker shoe. Without these changes, the contribution of this manuscript more lies in the lines 337-352 concerning the IMU based measurements of the upper body landmarks.

Response 17: We have further clarified the use of movement symmetry for assessing mechanical changes related to force production (and specifically asymmetries in force production between contralateral limbs). We are indicating that we are assessing the combined effect of trimming and re-shoeing. A previous study has utilized movement symmetry analysis to investigate trimming and re-shoeing.

Comment 18: This information can be of value to the industry if the manuscript is revised to a methodology paper in which the small sample size and lack of control would be less of an issue if this manuscript more focused on the methodology and avoided making any conclusions concerning the shoeing. 

Response 18: Thank you for your opinion on this matter. We respectfully disagree and believe that the new version of the manuscript addresses the main concerns relating to the ‘limb-by-limb’ protocol (of both limbs!) and the fact that we are presenting data for ‘re-shoeing’ and in relation to forelimbs.

Reviewer 3 Report

Comments and Suggestions for Authors

This paper presents an experimental study on the effect of re-shoeing with rolled rocker shoes on the movement symmetry of horses at trot measured by inertial sensor units. The study was performed on ten Warmblood horses trotting in straight line on hard and soft surface and lungeing. Significant changes were found on hard surface compared to soft surface  and in withers movement on circles. Results indicate increased push-off with the re-shod limb on the inside of the circle, and reduced weight-bearing of the re-shod limb and ipsilateral hind limb on hard ground.  

The strengths of this article lie in the quality of the protocol methodology (“limb-by-limb” shoeing protocol and analysis) and the well-founded and solid statistical analysis.

The article is well written, results are presented in a well-structured manner and discussed accurately.

I only found one typing error L242 – it’s not DPDup but DPDmax that is significantly affected by surface.

In conclusion, I find that this article brings valuable data on quantification of movement symmetry on horses after re-shoeing, especially lungeing on different track surface. This paper deserves to be published in Sensors.

Author Response

Comment 1: This paper presents an experimental study on the effect of re-shoeing with rolled rocker shoes on the movement symmetry of horses at trot measured by inertial sensor units. The study was performed on ten Warmblood horses trotting in straight line on hard and soft surface and lungeing. Significant changes were found on hard surface compared to soft surface  and in withers movement on circles. Results indicate increased push-off with the re-shod limb on the inside of the circle, and reduced weight-bearing of the re-shod limb and ipsilateral hind limb on hard ground.  The strengths of this article lie in the quality of the protocol methodology (“limb-by-limb” shoeing protocol and analysis) and the well-founded and solid statistical analysis.

Response 1: Thank you for your positive comments. We have in response to one of the other reviewers, who had expressed concerns about the ‘limb by limb’ protocol, further clarified the reasoning for this and have amended the title of the manuscript to include ‘limb-by-limb’.

Comment 2: The article is well written, results are presented in a well-structured manner and discussed accurately.

I only found one typing error L242 – it’s not DPDup but DPDmax that is significantly affected by surface.

Response 2: Thank you for spotting this mistake. We have corrected accordingly. Apologies for this oversight.

Comment 3: In conclusion, I find that this article brings valuable data on quantification of movement symmetry on horses after re-shoeing, especially lungeing on different track surface. This paper deserves to be published in Sensors.

Response 3: Thank you for your very positive evaluation of our manuscript.

Round 2

Reviewer 2 Report

Comments and Suggestions for Authors

While the authors did a good job of revising what was possible without redoing the study, there are several issues that need to be addressed. However, the manuscript has potential value to the farrier industry, and thus, authors are encouraged to continue with the suggested revisions. 

First off, the revised title is improved, but since there are kinematic differences associated with breed and performance type that has been well documented, authors need to include "trotting Warmblood show jumpers" to ensure this group is clearly targeted. With the exceptionally small sample size, the more targeted the focus of the study, the readers can avoid making potential incorrect assumptions. 

The introduction must prepare the reader with what is the bulk of the study, which is the limb-by-limb analysis. If previous work isn't available, then, this is an excellent point to bring up. Justifying why this work is being done needs to be clear at the beginning. Throughout much of the responses to the reviewer's comments the authors indicate the value of this approach, but this information must be in the introduction. Also, within the introduction further discussion concerning Warmblood show jumpers and potential forelimb issues that would require correct trimming and shoeing utilizing the rolled rocker toe shoe. This is hinted at in lines 63-70 but this needs to be further explored and supported with additional references.

For the methodology, a table is needed showing each horse's age, gender, weight, height, lameness assessment prior to the study (authors mentioned 'deemed fit' thus assuming this was through clinical evaluation), and training history (years and level of training). Further, due to the small sample size, authors need to include a power analysis showing the statistical validity of using only 10 horses. 

Further, a separate section labeled 'study limitations' needs to be added. The authors do mention some limitations throughout the discussion, but to organize the discussion better a separate section is warranted. Also, this limitation section should be developed with further discussion utilizing some of the comments made from the previous review that was unable to be address without redoing the study, i.e. lack of control, small sample size, potential asymmetry due to the limb-by-limb approach, etc. This will help future researchers to clearly see ways they can improve upon the current study and help to better organize the discussion.

Finally, with almost half of the references being a decade old if not older, while the manuscript tries to present new methodology utilizing new technology, this brings about concern that a thorough literature search was not completed. Authors are encouraged to do a more intensive review of literature to either replace or add to those references that are a decade old if not older. 
